# Sustainable Construction through the Lens of Neoliberal Governance: The Case of Vernacular Building Systems in Catalonia, Spain

Mónica Alcindor [1,2,*] and Delton Jackson [3]

1    CIAUD-UPT—Branch of CIAUD Research Center, Departamento Arquitetura e Multimédia Gallaecia, Universidade Portucalense Infante D. Henrique, Rua Dr. António Bernardino de Almeida, 541, 4200-072 Porto, Portugal
2    CIAUD, Research Centre for Architecture, Urbanism and Design, Lisbon School of Architecture, Universidade de Lisboa, Rua Sá Nogueira, Polo Universitário do Alto da Ajuda, 1349-063 Lisboa, Portugal
3    Studio UrbanArea LLP, Newcastle upon Tyne NE1 1EW, UK; delton@urbanarea.co.uk
*    Correspondence: monicaalcindor@upt.pt

**Abstract:** This paper asserts that neoliberal forms of governance are increasingly found in construction systems in Spain, a fact which becomes especially problematic when considering vernacular construction systems. Technological management and policy are both becoming more focussed on the promotion and consolidation of 'expert systems' at the expense of 'different' (and in particular) vernacular systems, which are processes which influence minds, and fundamentally shape subsequent actions. This paper adopts an ethnographic approach, undertaking investigation into the complexity of commonly found building systems, based upon empirical evidence gathered in the region of Catalonia. Focussing research on local vernacular construction systems reveals the extent to which the operation of distinct sets of managing 'technologies'—embedded in specific practices such as auditing—becomes instrumental in shaping local construction practices. Currently, locally distinctive practices are deeply impacted by social influences generated far away, which have the consequences of significantly influencing, diluting, or even erasing vernacular building systems, even where these represent an important source of sustainable building techniques.

**Keywords:** Spain; OCT; neoliberalism; vernacular building systems; construction governance; sustainability construction

## 1. Introduction

Governance—if understood in terms of explaining the exercise and establishment of political power—involves the control and regulation of a populace through multiple technologies and institutions in society [1]. Prior to this observation, the majority of studies of governance were commonly abstracted away from existing spaces and subjects. For this reason, they did not adequately engage with the ways in which people are constituted and ruled as neoliberal subjects through a multitude of 'technologies' and 'assemblages' of power, a perspective insightfully illustrated by theorists such as the philosopher Foucault [1,2]. Much progress has now been made, however, with governance being studied from a variety of different fields, from the management of agro-environments, to child minding and education.

Although there are authors who have analyzed architecture through the lens of neoliberalism [3–6], the study of building systems from this perspective has largely been overlooked, and a deep analysis is missing, together with the wider stories this can tell.

The study of building systems is similar to other research areas which makes it possible to observe governance taking place, but until now very little study has been undertaken in this specific area of practice. This applies particularly to vernacular building traditions,

which consist of specific types of building systems, but which have had little research into how relevant governance is applied (legislatively, regulatorily, or culturally), or into its consequences.

With regards to this, Rose and Miller [7] have usefully observed the relevance of knowledge and expertise for modern forms of governance, the extent of which cannot be overemphasised, especially where these are intrinsically linked to the administration of different kinds of construction, using (often competing) tactics which include education, inducement, incitement, encouragement, persuasion, and motivation.

Ultimately, these are concerned with certainty, with an interest in those technologies which aim to make reality "stable, mobile, comparable, combinable"; prerequisites which enable government to act upon it [7]. However, with regards to sustainability, these technologies induce universalist building systems which make it difficult to propose construction solutions linked to place, that is, those with low technology and/or low energy consumption [8,9].

Within this context, it is important to understand the concept of neoliberalism, as it defines the current era [10], yet it is neither objectively 'rational' or 'neutral'. In the study of governance—and its outcomes—the main focus of Foucault was to discover which kind of intrinsic rationality has been used, since political rationality is not a neutral form of knowledge but is instead an element of government that helps itself by creating a discursive field for framing thought and actions, in which exercising power is 'rational' [11].

However, Foucault rejected rudimentary 'capital logic' arguments on state-centred accounts and socio-economic development, with his analyses of discipline and governance attempting to explain the reasons behind economic exploitation and political domination [12]. His framework for interpretation and understanding investigated political strategies and the activities of authorities in their attempts to modulate decisions, actions, and events in the economy, the private firm, the family, and the behaviour and conduct of individuals [7].

To reiterate, this paper attempts to analyse neoliberal governance in building systems in Spain through an investigation into how management technologies have been applied, and their impacts on vernacular building systems in the region of Catalonia.

This paper is structured into four sections: firstly, a brief literature review of recent discussions on governance and neoliberalism is presented as foundation for the following sections. This is followed by a presentation of the methodology used in the research, then thirdly a presentation of management technologies applied to building systems in the specific case of Spain. The fourth section focuses on an analysis of the impact of this management technology on common vernacular building systems, particularly focussing on the specific case of Catalonia. The main conclusions are presented in the final section.

## 2. The Link between Neoliberalism and Construction Governance

### 2.1. Defining Concepts & Approaches

From the outset, it is important to establish definitions for key concepts and approaches used throughout this study, with regards to the main literature review, governance, and neoliberalism, and the ways in which these relate to the field of building systems.

The concept of governance is 'the regulation of conduct by the more or less rational application of the appropriate technical means' ([13], p. 106). Foucault approached this concept by focussing on the different meanings of conduct, both as 'personal conduct' and 'to conduct', or to be more precise, as 'the conduct of conduct', thus providing a term which ranges from 'governing the self', through to 'governing others'. That is to say, his efforts were focussed on showing how the modern state and the autonomous modern individual are entangled and co-dependent—what begins as an external directive is ultimately adopted as self-direction.

Governance is not normally conceived as a way to force people to comply with the will of the governor, but is instead a versatile equilibrium, balancing conflicts and complementarity between techniques that assure coercion, and processes through which the 'self' (of those governed) is constructed, modified, or controlled by itself [14]. Structuring

and shaping the field of possible actions of a populace effectively creates a cage without bars, through a heterogeneous array of regulatory practices and technologies, which end up as a reformulation of how to apply coercion or consensus, in which the latter is applied from 'autonomous' individuals' capacity for self-control [15].

Governance refers to the systemic, reflected, and regulated modes of power which go beyond the simple exercise of power over others, and include following specific forms of reasoning and rationality, which define either the telos of action, or the means to achieve it. Therefore, 'technologies' of government refer to the procedures, strategies, and techniques, through which different authorities seek to implement or enact programmes of government in relation to available forces and materials, and the oppositions and resistances anticipated or encountered [16].

Through the concept of governance, Foucault related technologies of being (involving common practices) with technologies of domination. This article aims to use the concept of governance to relate the construction decisions made by architects and other technicians with the technologies of domination analysed by the philosopher. The intention is to introduce how these management technologies also act in the field of construction since, as with other areas of human activity, they follow guidelines set in the foundations of the formation of the neoliberal state. Therefore, governance is a 'key notion' [15] one can use to understand the path followed by construction and building systems in the late modern period.

Nevertheless, it is important to point out that it is not government programmes or technologies which act, but rather the social forces deploying these programmes and technologies for their own particular purposes [17]. From this perspective, political programmes can be explained in terms of the underlying rationalities that shape their development [18], and in this respect it is tenable to suggest that auditing (introduced in the opening Abstract as a management technology which can be understood as a technology of domination) is at heart an ideologically driven system for controlling and disciplining architects, contractors, and so on (cf. [2]). According to Habermas, a certain form of hidden political dominance is imposed in the name of rationality ([19], p. 54).

In this regard, an understanding of neoliberalism is key in order to analyse the transformations of social practice and space which define the current era [10]. Specifically, the production and adoption of neoliberal mentalities and values regarding governance, especially attempts to enforce market logics in order to create conditions in which competition can flourish, and to depoliticise (through disempowerment, disenfranchisement, or delegitimisation) various social struggles over resources and rights [20]. It is within this context that this paper is presented, exploring and revealing the ideologically driven systems which operate through auditing in the field of construction, together with their profound effects.

Neoliberal rationalities comprise a number of coherent, ideologically driven political precepts pulled together by a fundamental belief in the superiority of free markets over intervention by mechanisms of the State [21], or the validity of its social responsibilities, concerns, or functions. Therefore, neoliberal forms of governing attempt to extend market relations into every domain [22]. The increasing dominance of market instruments (or more broadly speaking, the 'market') over the governance and control of construction systems is a characteristic feature of what this paper identifies as the "neoliberalisation of building systems". In relation to 'building systems management', market instruments may be defined as those initiatives that 'aim to mobilize individual incentives in favour of positive outcomes', through a careful modulation and calculation of benefits and costs associated with the strategies of particular building systems. At the same time, it is important to question what is meant by 'positive outcomes', if sustainability (for example) has not been taken into account as part of the equation [23].

Neoliberal forms of governance are typically viewed as colonisation of the social, through processes of deregulation, marketisation, and privatisation, in which the state takes a minimal role [24], in spite of an increasing recognition of the importance of the role

played by state agencies in enabling markets to work efficiently [25]. Critically, neoliberal forms of governance enable technical expertise to present an appearance of addressing safety in construction (for instance), while simultaneously creating and securing conditions for further capitalist accumulation (cf. [26]), and the achievement of narrowly defined ends which include financial gain, and/or positioning into power relationships. Today, domination is perpetuated and extended not only through traditional routes such as regulation, policy, or enforcement, but through technology as well, as this avenue provides the discrete legitimisation of an expansive political power that permeates and engulfs all areas of culture, including construction [19]. Until now, however, little attention has been given to the consequences of the neoliberalisation of building systems.

### 2.2. Relevance of Neoliberal Governance in Building Systems

The relevance (as well as the potential contribution) of the concept of neoliberal governance in building systems can be seen more clearly with regards to three main areas that intertwine with each other:

1. Concept of 'political knowledge'. Foucault appears to offer a useful and important way for understanding the relationship between governmental practices and territories, in particular how places are governed and shaped, in ways which are ostensibly subject to mathematical modelling and control [27], steering rather than dictating through processes of abstraction and simplification [28]. There is more to the process of state spatialization, however, than simple policing or repression, and it may be more important to look at the multiple, less dramatic, and mundane domains of bureaucratic practice, through which states reproduce scalar hierarchies and spatial orders. In other words, it is the 'know-how' or the practices which make government possible [7], including management technologies. In particular, it is the technology of efficiency which has transformed administration into bureaucracy, technologies upon which bureaucracies now depend [29]. Furthermore, it must be considered that states are not simply functional, bureaucratic, or mechanistic organisations, but also powerful centres of symbolic and cultural production [30].

2. Concept of market independence from state affairs. Foucault highlighted that the power of the economy rests on a previous 'power economy', since the accumulation of capital implies forms of work and production technologies that allow the use of multitudes of human beings in economically profitable ways. Foucault located strategy not in actors but in clear controls, which, in turn, are the outcome of, rather than a condition or determinants of, the dynamics in local settings, where microphysics of power continuously create new relationships between knowledge and the exercise of power [28].

3. Domination and technologies of the self: developing indirect techniques to lead and control individuals. Government is historically the matrix which articulates the dreams, strategies, manoeuvres, and schemes of authorities, seeking to shape the conduct and beliefs of others in desired directions, by acting upon their circumstances, their will, or their environment [7]. One key feature of neoliberal rationality is the congruence it works to create between the idea of a responsible and moral individual, and an economic-rational individual—that moral responsibility is somehow intertwined with and subject to fiscal imperatives and accountability.

In the field of building systems, auditing is an essential part of the 'new public management', as it highlights and stresses the 'control of control' through a characteristic focus on the reliability and effectiveness of expert systems, which are 'systems of technical accomplishment or professional expertise that organise large areas of the material and social environments in which we live today' ([31], p. 27). At the same time, expert systems rely heavily upon a 'power economy' characterised by well-established distribution and marketing processes which facilitate their imposition. These are further assisted by offering clear financial value chains (and excluding wider social or environmental accounting), driven by a neoliberal perspective that moral responsibility is indistinguishable from economic

rationality. Therefore, audit management enables the effective functioning of a dispersed and decentralised state in controlling the construction activities of an individualised public, through mundane bureaucratic processes subject to mathematical modelling.

The task then is to draw attention to the creative and social processes, through which a state hierarchy becomes effective and authoritative in the field of building systems, and how this affects the sustainability of vernacular building systems in particular.

Vernacular building systems, until well into the modern age, have followed principles of tradition anchored to place. The predominant source of their organisation and construction was the established order of traditional society. Therefore, vernacular building systems are a critical field of study, where the profound effects of new (mundane and bureaucratic) practices can be easily seen, even though they often slide unnoticed and unremarked below the threshold of discourse. Where new systems of auditing become established, however, they bring the risk of damaging local cultures of first-order practice and its sustainable characteristics. To fully examine and interpret this practice-oriented concept an ethnographic approach is required.

### 3. Methodology

In conducting the research, an interpretivist paradigm has been used, conscious that the patterns (and associated data) sought and found in interpretations of social reality are not immutable, or 'laws' in the sense given to them by positivist sociology. Since there is no separation between the observer and the reality being studied, knowledge is produced from understanding. In other words, the key points of this research consist of situating itself in the perspective of all the participants in the construction process, the importance of the context, and the holistic and processual evaluation of the object of study, renouncing the imposition of closed hypotheses from the outset [32] by pursuing an inductive path of investigation. Therefore, an in-depth analysis and study of the cultural dimension has been favoured over simply parsing quantitative data, providing a new filter for understanding not only the technical approach used in the vernacular building system, but also the forces influencing construction technology more generally as well.

The nature of this study does not attempt to be conclusive. Instead, it seeks to explore and discuss ideas for progressing the academic study of sustainability, and the need to study building systems from a social perspective. For this reason, the anthropology of building systems has been the focus, and an interpretivist paradigm appropriate to the nature of the subject has been used (coupled with a qualitative methodology), identifying how governance through management technologies affects the sustainability of traditional building systems. This has required an exploration of the relationships between communities, building systems, and neoliberal governance. This is most accurately achieved (according to [33]) by means of the ethnographic method, which helps to identify underlying causes, while attempting to address the complexities involved by studying relationships between micro-level behaviours and macro-level phenomena.

This study does not attempt to be definitive or an end in itself, but instead to identify trends and potential relationships between variables in a way which invites and signposts avenues for further research.

As stated, an ethnographic approach has been used for this research, based on three main elements. This has included 63 semi-structured interviews with relevant individuals (within the context of refurbishment), specifically comprising 21 builders, 28 architects, and 14 masons, materials distributors, and other professionals. The principle used for conducting the interviews was based upon the 'saturation of the sample', which is to say that interviews were conducted until the answers became repetitive.

Participant observation of work sites in Catalonia has also been another key tool, and the professional experience of the researchers allowed a close knowledge and understanding of the activities of the agents involved. This also enabled a deeper investigation into the complexity of the most commonly used construction solutions.

On this basis a representational account of the interactions between actors and processes (operating on diverse spatial scales) will be attempted, together with the ways in which these interactions eventually emerge into specific building systems.

## 4. Management Technologies in the Construction Systems Field in Spain

On 6 May 2000 the Spanish Building Ordinance Law came into force [34]. As a result, it was the developer who became responsible for construction insurance. Therefore, at this time, the audit processes developed by insurance companies were also indirectly established by the new law. Insurance Companies in Spain, most of them grouped in the Spanish Union of Insurance and Reinsurance Entities (UNESPA), try to develop technical documents approximating a real risk assessment. As a basis, these are tied to Decennial Insurance, Spain's only compulsory insurance, which offers ten years' cover for any defects in construction.

It may be that an institutional lack of faith in architects and technicians related to construction issues has led to the emergence of an oversight industry, in order to satisfy a demand for signals of order. Regardless, the key point to be understood is that 'any' level of risk is now considered unacceptable; risk must be avoided at all costs [35]. It is what Amoore and de Goede [36] named precautionary risk, the 'risk beyond risk'.

Claim statistics recognise that 43% of these risks are due to project errors, 30% to poor execution, 15% to material defect, and 8% to lack of proper maintenance, with the remaining claims due to other factors [37]. Given the importance of the risks, the immediate approach of the insurance companies was to find the right people or organisations to carry out the inspection and technical assessment work; that is, those with sufficient knowledge, responsibility, and independence to support the insurance offered. This was carried out through the performance of recognised expert technicians [37], and the establishment of a definition and control system for the different construction processes. The creation of a company with the necessary economic solvency that could take on this new task was also necessary, hence the emergence of the so-called Technical Control Organisations (OCT—Organismo de Control Técnico). In order to qualify for ten-year insurance cover, an OCT must be hired, which will be in charge of the technical control of the work, and for issuing a series of essential reports before the ten-year insurance can be obtained, and which address three points: project control, execution control, and control of trials.

The control of the project assesses the rationale(s) for the chosen construction solutions, the adequate definition for a correct execution, the qualities and characteristics of the different elements, as well as an adequate and correct definition of the budget. The control of the execution consists in verifying that it is carried out following the definition established in the project, the current regulations, as well as the technical knowledge sanctioned by practice. The control of tests verifies the follow-up of the quality control plan, and the suitability of the tests carried out, as well as the request for new ones if necessary.

At the same time, there are three basic criteria to be met by OCT technical agents: independence, technical expertise, and non-biased assessment. Independence is guaranteed with the absence of conflicts of interest with the works being audited. On the other hand, non-bias does not yet have a defined method of control. Technical expertise must be accredited, but in Spain this has been difficult to control due to the official absence of this as a recognised field of activity [38]. This fact has initially caused other control processes to be initiated and implemented by the OCT themselves, in order to ensure correct and consistent standards based on the experience of technicians and types of work to be audited, and the volume and height of the work, as well as the type of terrain, and its construction characteristics. Behind this, there is the principle that regulatory systems increasingly rely upon the 'control of control' [2].

OCT is the technology of government implemented in the building field, through which political rationalities become capable of deployment. In this way, the complex assemblage of diverse forces comes to be regulated by authoritative criteria through mundane

mechanisms that enable rule 'at a distance'. Since 2000, the auditing of this internal control for self-checking arrangements has continued to grow as an industry (cf. [39]).

## 5. How Management Technology (OCT) Affects Vernacular Construction Systems

This area of study focussed on one of the regions of Old Catalonia (Catalunya Vella), the Baix Empordà.

The transformation of the practices in vernacular building systems which the OCT mandated can be evidenced by analysing three particular building systems of the region: the tile vaults, the structural use of ordinary masonry walls, and the use of local wood species.

### 5.1. Construction of Tile Vaults

The traditional tile vault has three defining characteristics: construction without formwork, the use of gypsum paste or plaster as a binder, and the use of brick (Figure 1).

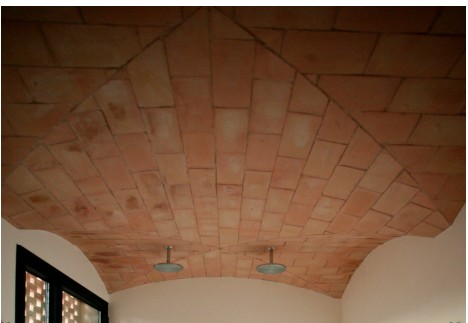

**Figure 1.** Traditional tile vault.

Traditionally, the construction of the first layer was the most delicate stage, since it required the most mastery and skill. The binding material was gypsum due to its rapid setting; an important factor, as this is what made it possible to dispense with formwork in most cases. The low weight of the ceramic tiles or bricks was also an important consideration, and once the first layer was complete, additional layers (if any) could be easily added to increase strength, or to provide aesthetically pleasing finishes.

With the entry of the OCT, the continued use of this vernacular construction system depended on validation through mathematical tools, to model future behaviour in the face of different situations throughout its useful life. Despite centuries of use, insurers demanded the introduction of a layer of concrete, which would provide the security afforded by a predictive mathematical model that is associated with this material. Up until that point the tile vault was anchored to tradition, one of the legitimate bases of management of the actions of community life in the past.

OCT have also become influential agents of change, since ways of understanding human progress have increasingly come to be governed by trust in and reliance upon the scientific-technical domain, with less consideration given to the empirical knowledge of artisans and communities accrued over time [40]. This is further compounded by neoliberal rationalities which have created conditions in which various social struggles over resources and rights have been depoliticised, and communities become disconnected from historic practices and perspectives, including ideas of stewardship. The benefits of traditional systems with proven track records over time are not taken into account, such as the use of materials with low embodied energy, rooted in an integrated approach to practice, and strictly linked to a conception of the world based on the careful management of local resources [41].

### 5.2. Structural Use of Ordinary Masonry Walls

Walls traditionally fulfilled two specific functions: as a structural element, supporting either wooden beams or vaults; and as room dividers (exterior and interior), with both of

these functions reliant on the thickness of the walls themselves. With the appearance of OCT, however, traditionally constructed walls were no longer considered able to fulfil a structural function, even in the refurbishment of vernacular buildings—a situation which effectively illustrates the extent to which governance can structure and shape the field of possible actions of subjects. Instead, walls became relegated to use only as envelopes, in favour of new structural techniques, materials, and practices which could be validated using mathematical models predictive of future behaviour. However, these changes often bring with them greater environmental impacts and energy costs, both in obtaining the material, and for future recycling, while at the same time creating and securing conditions for further capitalist accumulation (as noted earlier) (Figure 2).

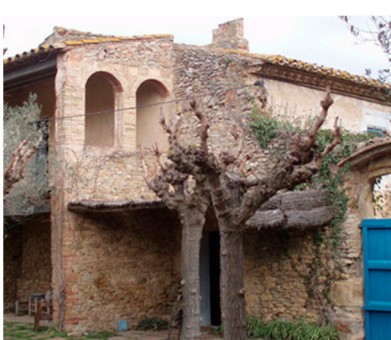

**Figure 2.** Typical (traditionally constructed) masonry walls.

### 5.3. Use of Local Wood Species

There are around 30,000 different species of wood in the world, but only 2000 of these are used commercially, and of the 150 of these marketed in Spain, of which only a few are used for structural purposes.

Traditionally, both white and black poplars were used for construction in the area of Baix Empordà, however, Spanish regulations consider black poplar unsuitable for structural use, and it is no longer used for creating roof beams. Therefore, it is very difficult to build using local woods [42]. Instead, the commercially available (and approved) wood comes from the forests and wetlands of Northern Europe, which are better adapted to existing technologies for efficient cutting and harvesting. At the same time, the faster growth rates and smaller dimensions make this timber largely unsuitable for use as structural beams, which has led to its extensive use in laminated timber, that is, wood cut into small pieces of homogeneous length and joined with resins. A higher reliability and predictability are cited as the reasons for the preferred use of laminated timber, and since the preferences of architects are shaped through the effects of social forces (including education, knowledge, budgets, personal liability, and so on), this does not preclude them from intervening creatively to transform existing social structures (cf. [43]) which in turn affect wider attitudes. There is also a belief held (and promoted) that pathologies caused by hygroscopic movements are lower for laminated timbers, as are the risks of biotic attacks.

It is mainly this predictability factor—promoted by the OCTs—that makes this type of wood more favourable over local solid natural wood, which traditionally defines the character of roof construction in the region. In summary, what lies behind this preference is the legitimacy achieved by architects as 'objectifiers of chance', reducing construction (as well as professional and budgetary) risks, whilst exercising not only an ability to reject traditional alternatives, but to erase them as well [44]. One of the principles of sustainable construction is the favoured use of local materials with minimal entropy and closed lifecycles [45], however, these end up becoming a less viable option as a result. This is accompanied by inducements of further capitalist accumulation, supported by neoliberal forms of governance and an appearance or guise of addressing safety in construction.

Although OCTs have a tightly defined regulatory function, rather than holding opinions on the ideology of which solutions become adopted or executed, OCT for audit inevitably force changes in building systems' practices [44]. These effects end up being

systematically recorded and iteratively fed back into the design process, and because audits do not operate neutrally, they end up having effects on the audited. Structuring and shaping the field of possible action of architects and technicians effectively creates an unacknowledged and invisible force modulating the 'autonomous' individual's capacity for self-regulation. Eventually, architects and technicians themselves change or reject the traditional vernacular construction systems which require acceptance by OCT (Figure 3).

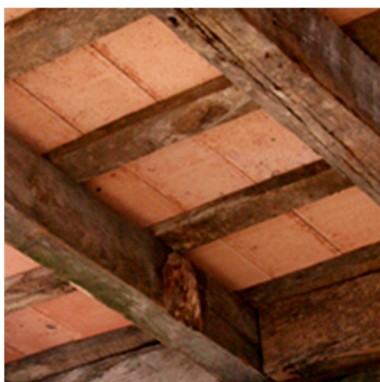

**Figure 3.** Wooden structure.

## 6. Final Considerations

The impacts of new systems of auditing upon vernacular building systems is a critical field of study, as they bring the risk of damaging local cultures of first-order practice and associated sustainable characteristics.

The task has been to draw attention to the creative and social processes through which a state hierarchy becomes effective and authoritative in the field of building systems, and how this affects the sustainability of vernacular building systems in particular.

Decennial insurance issued by the OCT is the main governmental technology which has led to the dismantling of traditional forms of construction in Spain. OCT is the unquestionable authority mandating a precautionary approach to risk reduction, which in turn destabilises and reshapes the basis of vernacular construction systems, since the risk–security complex inadvertently, but effectively, puts managerial technology in the driving seat. Generally, security and liberty are viewed as forming a zero-sum game, so measures of security may be used as justification to occasion a reduction of a technician's individual liberty (c.f. [35]) since there is a close relationship between risk rationalities and 'targeted governance' [46].

Quality assurance programmes require the establishment of aims, objectives, and design performance criteria, as well as measures for implementation and delivery. Therefore, any departure from contemporary norms which involve the use of vernacular construction systems involve two risks. The first involves the higher costs associated with the non-standard (non-standardised) or bespoke construction which historic solutions have come to represent. At the same time, these solutions must also be justified by extraordinary trials and tests to ensure their safety and durability for each specific project. Within this context, it is also important to consider that neoliberal forms of governing do not take the existence of oligopolies into consideration, which compete under unequal conditions.

Secondly, there is a burden placed upon the technician, who must assume the responsibility for not using solutions recommended by the various different regulations. In this case, the difficulty becomes even greater, since the effectiveness of traditional techniques depends on interactions between many factors which must be taken into careful consideration. To be successful, vernacular architecture must follow the principle of tradition anchored to place and requires an understanding of both materials and building systems, together with how these relate to the site and context.

Under these conditions, even well-established and locally distinctive traditions are deeply affected and reconfigured by external social influences that have been made blindly

at a distance. As a result, vernacular construction systems have become diluted, if not erased, as neoliberal government practices (born from and supported by rational–legal systems), manage to penetrate to differing extents to the very heart of the local level [44]. Auditing has unwittingly been introduced as an agent of change without a measured consideration of benefits relative to any possible dysfunctional effects—risk assessments effectively aimed at protecting capital, but without similar regard given to social or environmental protections, since auditing (and related ideas of monitoring) are uncritically understood as 'positive' measures, within the paradigm of safety and security as the normal concern and purview of government. Thus, it is argued that neoliberal government abdicates its responsibility when it comes to delivering public goods (including benefits or services), by denying (or even destroying) them under the directives of related management technologies, as illustrated by the subjugation of vernacular building systems to technological systems instead (c.f. [29], p. 19). For this reason, the use of vernacular building systems requires tailored policy, giving full consideration to triple-bottom line benefits and disbenefits, within the context of community, identity, and environment. As Elionor Ostrom [47] pointed out, increasing the authority of local people and communities to devise their own rules, may well result in processes which allow vernacular building systems to flourish and evolve, taking sustainability into account and solving problems through collective action.

**Author Contributions:** Conceptualization, M.A.; methodology, M.A.; analysis, M.A. and D.J.; investigation, M.A.; writing—original draft preparation, M.A.; writing—review and editing, D.J. All authors have read and agreed to the published version of the manuscript.

**Funding:** This work is financed by national funds through FCT—Fundação para a Ciência e a Tecnologia, I.P., under the Strategic Project with the references UIDB/04008/2020 and UIDP/04008/2020.

**Informed Consent Statement:** Informed consent was obtained from all subjects involved in the study.

**Data Availability Statement:** No new data were created.

**Acknowledgments:** We would like to acknowledge all the key informants for their contribution, particularly Olalla Rios.

**Conflicts of Interest:** The authors declare no conflict of interest.

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
