# Peer review of "Sustainable Construction through the Lens of Neoliberal Governance: The Case of Vernacular Building Systems in Catalonia, Spain"

_sustainability, doi:10.3390/su151813812_

Round 1

Reviewer 1 Report

Dear the authors,

Thank you for the opportunity to review your paper.

This paper is at the level of a report

1. This paper presents an interesting and potentially publishable piece of research that would be of interest to both the wider academic research community as well as stakeholder in the construction industry.

However, this study shows general information. Therefore, there is not enough reason for this paper to be conducted. And the originality of this study is not enough.

2. This paper is unclear and complex. This paper is difficult to understand due to the complicated flow of research.

3. This paper is written only with unclear concepts. It should be rewritten with examples that can be applied in practice.

4. For literature review of recent discussions on governance and neo-liberalism, it should be specifically presented which topics were discussed. The study should show a definition of neo-liberalism. Research should show an exact concept of the relevance of neo-liberal governmentality in building systems using some figures. In addition, a sufficient literature review should be conducted.

5. Statistical analysis should be performed on the information obtained through the interviews.

6. In Lines 226-227, Organismo de Control Técnico must be edited in English.

In Lines 226-227, Organismo de Control Técnico must be edited in English.

Unnecessary words appear on several sentences.

Author Response

Thank you for taking the time to review this paper, and for the comments provided.

In the present revision, the following modifications have been made to the text of the article to try to introduce the improvements that the referee is suggesting.

The criticism of being an unclear and complex paper has been tried to improve the flow by changing the order of some paragraphs, adding new sentences and being more precise in the use of keywords such as governance and government.

The key concepts that structure the theoretical framework are mainly governance and neoliberalism and to enhance the practical understanding, it is possible to find it in section 5 “How management technology (OCT) affects vernacular constructions systems new statements”. This section establishes the link between them. Also, it has reinforced the reasoning.

It is possible to find a definition of neoliberalism from the line 147 to 213.

The intention of this article is to highlight the “embedding” of constructive action in cultural conditions, specifically neoliberal governance. The anthropology of building systems has been the focus and an interpretivist paradigm appropriate to the nature of the subject has been used, coupled with a qualitative methodology, which attempts to address the complexities involved by studying relationships between micro-level behaviours, and macro-level phenomena. Therefore, the positivist, quantitative approach has been avoided, in favour of an in-depth analysis and study of the cultural dimension, providing a new filter for understanding the forces influencing construction technology more generally as well. Perhaps a workable solution would be a statement to this effect in the methodology section (regarding the interpretivist paradigm), to provide clarity for readers about the qualitative methodology used.

The abbreviations of the Technical Control Organisations are only used in Spain, which is why it has not been considered necessary to translate them into English as they will not be recognised beyond Spanish territory.

With many thanks and Best regards,

M. Alcindor & D. Jackson

Author Response

Thank you for taking the time to review this paper, and for the comments provided.

In the present revision, the following modifications have been made to the text of the article to try to introduce the improvements that the referee is suggesting.

As the referee has pointed out the neoliberalism analysis in architecture is not a new area, so it has been introduced with the bibliographies that support this fact, and it has been highlighted what the new field is intended to face, the neoliberalism analyses of building systems.

It has been more careful in avoiding confusion between similar terms, such as governance and government.

It has reinforced the neoliberal approach by referencing Habermas, Ellul and Ostrom’s works.

With many thanks and Best regards,

M. Alcindor & D. Jackson

Reviewer 3 Report

The article is an interesting – even if a little synthetic – contribution to the research on the relationship between the political setting of the construction sector and the conservation of the vernacular heritage; the article is properly structured and well written.

The article focuses on the Spanish case, but neglects at least in part the more general picture of international research. Consequentially the list of references lacks some contributions that are significant to the topic covered in the article, especially those related to the application of a lifecycle approach for the sustainability assessment and preservation of widespread and vernacular built heritage, others on the risks for preservation of vernacular built heritage deriving from culturally unfounded recovery interventions as well as from non-intervention (also deriving from a lack of regulation or from poorly set regulations), and others on people education and participation in preservation of vernacular heritage.

As a minor comment, I may add that some words were capitalized when it would not have been necessary (e.g., the word «Developing» that follows a colon in line 175, and the word «That» in line 181, that follows a long or em dash; in addition is not clear why the authors put the dash in that sentence).

Author Response

Thank you for taking the time to review this paper, and for the comments provided.

The bulk of the literature has been focused on governance issues addressing the poorly set regulations to deal with traditional building systems. We think that the introduction of a deeper literature focus on sustainable assessment or other issues related to vernacular built heritage (risks linked to interventions or non-interventions, citizens participation and education) carries the risk of scattering the thread of argument of the effects that management technologies have on the use of traditional building systems.

The following references are about sustainability and vernacular built heritage:

  1. Cuchí Burgos, A. (2005). Arquitectura i sostenibilitat: TTS, Ediciones UPC: Barcelona.
  2. Asquith, L., & Vellinga, M. (Eds.) (2005). Vernacular architecture in the twenty-first century. Taylor &Francis
  3. Vellinga, M. (2015). Vernacular architecture and sustainability: Two or three lessons. In Vernacular Architecture: Towards a sustainable Future, eds. C. Mileto, F. Vegas, L. García Soriano, V. Cristini, Taylor & Francis Group, London, pp.3-8.
  4. Naredo, J. M. (2010). Raíces económicas del deterioro ecológico y social. Siglo XXI de España Editores, SA.
  5. Laureano, P. (1999). Agua: el ciclo de la vida. Barcelona: Naciones Unidas: Agbar: CCD, DL.
  6. Alcindor, M., & Coq-Huelva, D. (2020). Refurbishment, vernacular architecture and invented traditions: the case of the Empordanet (Catalonia). International Journal of Heritage Studies, 26(7), pp. 684-699.
  7. Alcindor, M. (2019). “Locations of the Global in Traditional Architecture.” In ICOMOS -CIAV&ISCEAH 2019 Joint Annual Meeting & International Conference on Vernacular & Earthen Architecture towards Local Development, Tongji University Press, Pingyao, China, pp. 351–357.
  8. Braungart, M., & McDonough, W. (2009). Cradle to cradle. Random House

Minors comments have been fixed.

With many thanks and Best regards,

M. Alcindor & D. Jackson

Reviewer 4 Report

This study presented research results through accumulated data such as fundamental causes and responses to Spain's construction system through empirical evidence for the Catalonia region. 1. Regarding the historical development of neoliberalism and construction governance, it is necessary to organize them around the scope and main meaning, and avoid listing existing studies. 2. In particular, it is necessary to improve the understanding of how the premise of the researcher's conceptual definition analysis of the relationship to neoliberal governance within the building system was deduced. 3. There is a lack of organic connection between chapters 3 and 4. Relatively, a sequential explanation of diagramming and analysis techniques is needed rather than a descriptive description of the methodology in Chapter 4. 4. For the conclusion, a brief summary of the background and purpose of the study, the premise of the analysis, the conclusion drawn, and the implications and limitations thereof should be sufficiently presented.

It is necessary to supplement academic terms and expressions.

Author Response

Thank you for taking the time to review this paper, and for the comments provided.

In the present revision, the following modifications have been made to the text of the article in order to try to introduce the improvements that the referee is suggesting.

It has improved the understanding of neoliberal governance in building systems in lines 316 – 323. But also adding some sentences in order to enable this understanding.

To solve the lack of connection between chapters 3 and 4, they have been changed the order of them.

It has been enhanced the methodology section introducing a more extensive explanation of its nature.

In the conclusion section, it has been structured as it was suggested.

With many thanks and Best regards,

M. Alcindor & D. Jackson

Round 2

Reviewer 1 Report

Dear the authors,

Thank you for the opportunity to review your paper.

This paper is still at the level of a report

This study only shows surveys and findings, and no specific research was conducted. Additional research is needed based on this survey.

1. The objective of this paper is still unclear and complex. This paper is difficult to understand due to the complicated flow of research. Therefore, it is necessary to revise the overall structure of this paper including the title of each chapter.

2. This paper is written only with unclear concepts. It should be rewritten with examples that can be applied in practice.

3. In the Introduction, the flow of the study should be added as a figure, and explanation is required. In accordance with the flow of this study, overall revisions should be made systematically and logically. If this is not fully explained, it cannot be accepted.

4. What are the criteria for distinction and differences between the chapter 2 and 2.1?

Whole manuscript should be completely rewritten in order to upgrade it to cope with the international academic standard.

Author Response

Dear Reviewer,

Thank you very much for your comments, which we have addressed as far as possible, in order to improve the paper in line with your suggestions. However, we would appreciate further specific guidance, in particular for those comments which are general in nature.

We would also like to take the opportunity to provide further clarification on the content and intent of the paper, and the theoretical framework and methodology which we’ve used.

The thread of the article has been elaborated through the presentation of the literature dealing with the two main concepts upon which the article is structured, namely governance and neoliberalism. In section 2.1, care has been taken to clearly define the concepts used, by taking into account the main authors of reference on these issues. The chapter has been divided into two sections, one more focused on the definition of the concepts, and a second focused on the specific aspects that relate general theory to the specific field of building systems. Therefore, following the suggestions made, a new title has been introduced to clarify the difference between chapters 2 and 2.1.

The methodology used to investigate these issues (and the reason for the choice of methodology) is then presented. From lines 240 to 264 an attempt has been made to explain the paradigm behind the ethnographic research employed and its specificities, as well as the reasons why this methodology is appropriate for the phenomenon under study. The review comments are critical however, but are unclear on how these lines should be improved, and we would welcome further guidance. The rest of the chapter defines the theoretical sample size based on `saturation of the sample’, a well-established criterion used by the anthropological community.

After this, the management technologies employed in Spain were presented, providing the context for understanding how the theoretical framework from Chapter 2 is embodied in the examples provided in the subsequent chapter, "How management technology (OCT) affects vernacular construction systems".

The suggestion of a total review however presents a difficulty, as this unfortunately conflicts with the other three reviewers who have accepted the paper, subject to minor revisions. In this instance we feel it best to defer to the academic editors to make a decision on this issue - To maintain a consistent approach which respects the views expressed by all the reviewers, addressing their concerns where possible, and establishing a framework for any compromises which might be necessary to resolve differences of thought or opinion. We hope you understand our position, and thank you again for your insights, time, and advice in reviewing our paper.

Best regards,

M. Alcindor & D. Jackson

Reviewer 2 Report

Thank you for thoroughly responding to my comments and concerns. I think the paper is much stronger now, and suitable for publication. There are a few places where the text needs minor editing, and I would again caution the authors to be sure the distinction between "governance" and "government" is always clear. Otherwise, I congratulate you on a good job.

As noted above.

Author Response

Dear Reviewer,

Thank you very much for your comments, which we appreciate for helping to improve our paper.

With many thanks and Best regards,

M. Alcindor & D. Jackson

Reviewer 4 Report

First of all, I think that the manuscript was supplemented intensively in a short period of time. In addition, it is judged that the implications of related topics in realistic cases have been met at an appropriate level. However, I cannot agree with the simplification of methodological discussion, but I expect that many supplements and reflections will be made in other studies in the future.

Appropriate corrections have been made.

Author Response

(The authors gave the same response as above.)

Round 3

Reviewer 1 Report

Dear the authors,

In the Methodology, Reasons for conducting the research and a logical basis for the methodology are necessary. It should be explained more convincingly.

English needs to be calibrated. Unnecessary words appear on several sentences.

Author Response

Dear Reviewer,

Thank you very much for your comments, which we have addressed as far as possible, in order to improve the paper in line with your suggestions.

A paragraph has been introduced in the methodology section emphasising the initial nature of this work, and its importance in opening up new avenues of research that will involve the use of different methodologies.

Best regards,

M. Alcindor & D. Jackson
